# Epigenetic Profiling in the Saliva of Obese Pregnant Women

**DOI:** 10.3390/nu14102122

**Published:** 2022-05-19

**Authors:** Chiara Mandò, Silvio Abati, Gaia Maria Anelli, Chiara Favero, Anaïs Serati, Laura Dioni, Marta Zambon, Benedetta Albetti, Valentina Bollati, Irene Cetin

**Affiliations:** 1Department of Biomedical and Clinical Sciences, Università degli Studi di Milano, 20157 Milan, Italy; gaia.anelli@guest.unimi.it (G.M.A.); anais.serati@unimi.it (A.S.); irene.cetin@unimi.it (I.C.); 2Department of Dentistry, University Vita-Salute San Raffaele, 20132 Milan, Italy; abati.silvio@hsr.it; 3EPIGET LAB, Department of Clinical Sciences and Community Health, Università degli Studi di Milano, 20122 Milan, Italy; chiara.favero@unimi.it (C.F.); laura.dioni@unimi.it (L.D.); benedetta.albetti@unimi.it (B.A.); valentina.bollati@unimi.it (V.B.); 4Department of Pathophysiology and Transplantation, Università degli Studi di Milano, 20054 Segrate, Italy; 5Department of Woman, Mother and Child, Luigi Sacco and Vittore Buzzi Children Hospital, ASST Fatebenefratelli-Sacco, 20154 Milan, Italy; martazambi@hotmail.com; 6Occupational Health Unit, Fondazione IRCCS Ca’ Granda Ospedale Maggiore Policlinico, 20122 Milan, Italy

**Keywords:** pregnancy, maternal obesity, GDM, oxidative stress, inflammation, epigenetics, miRNA, DNA methylation, saliva, periodontal disease

## Abstract

Maternal obesity is associated with inflammation and oxidative stress, strongly impacting the intrauterine environment with detrimental consequences for both mother and offspring. The saliva is a non-invasive biofluid reflecting both local and systemic health status. This observational study aimed to profile the epigenetic signature in the saliva of Obese (OB) and Normal-Weight (NW) pregnant women. Sixteen NW and sixteen OB Caucasian women with singleton spontaneous pregnancies were enrolled. microRNAs were quantified by the OpenArray Platform. The promoter region methylation of Suppressor of Cytokine Signaling 3 (*SOCS3*) and Transforming Growth Factor Beta 1 (*TGF-Beta1*) was assessed by pyrosequencing. There were 754 microRNAs evaluated: 20 microRNAs resulted in being differentially expressed between OB and NW. microRNA pathway enrichment analysis showed a significant association with the TGF-Beta signaling pathway (miTALOS) and with fatty acids biosynthesis/metabolism, lysine degradation, and ECM–receptor interaction pathways (DIANA–miRPath). Both *SOCS3* and *TGF-Beta1* were significantly down-methylated in OB vs. NW. These results help to clarify impaired mechanisms involved in obesity and pave the way for the understanding of specific damaged pathways. The characterization of the epigenetic profile in saliva of pregnant women can represent a promising tool for the identification of obesity-related altered mechanisms and of possible biomarkers for early diagnosis and treatment of pregnancy-adverse conditions.

## 1. Introduction

The obesity epidemic is expected to involve almost 21% of women all over the world by 2025 [1], representing a serious risk for women of reproductive age. Maternal obesity during pregnancy represents a danger for short- and long-term health outcomes for both mothers and children [2,3,4]. Indeed, up to 50% of obese pregnant women develop Gestational Diabetes Mellitus (GDM) [5], and the offspring of obese and diabetic mothers are at higher risk of developing metabolic and cardiovascular diseases in early childhood and later in their adulthood [6], with possible transgenerational effects [7]. Moreover, newborns of obese and diabetic mothers have been recently shown to be hypoxic, acidemic, and to have increased oxidative markers compared to normal-weight pregnancies [8,9].

The metabolic and hormonal balance is dysregulated in obesity and diabetes because of excessive adipose tissue deposition and intracellular fatty acids, leading to alterations of the immune cell profile and inflammation [10]. This metabolic-dependent pro-inflammatory background increases oxidative stress and insulin resistance, which further contribute to metabolic dysregulation, in a deleterious vicious circle. Pregnancy itself is characterized by a physiological generalized low-grade inflammation, and maternal obesity can contribute to additional lipotoxicity, inflammation, and oxidative stress, which may strongly impact the intrauterine environment, with significant effects on fetal health and metabolism [8,11,12]. We previously reported alterations in the placentas of obese mothers, showing oxidative stress and a dysregulated placental metabolome [13,14,15]. These data suggest that obesity-related increased inflammation and oxidative stress may determine a cascade of events, leading to systemic and local dysfunctions and consequent impaired pregnancy outcomes.

Among obesity-related recurring comorbidities, periodontal diseases represent an additional important risk factor for pregnancy outcomes [16]. During pregnancy, both obesity and diabetes can cause oral dysbiosis [17], which is the primary cause of periodontal diseases [18]. In turn, maternal periodontal diseases have been associated with poor obstetric outcomes, such as pre-term birth, premature rupture of membranes, and low birth weight, especially in obese women with pre-gestational obesity and GDM. The systemic inflammatory alterations induced by both obesity/GDM and periodontal diseases can indeed activate adverse responses at the maternal–fetal interface [19,20,21,22]. We recently reported increased total antioxidant capacity and inflammatory levels in both the saliva and blood of obese pregnant women with and without GDM, suggesting a synergic detrimental effect of obesity and periodontal diseases [23].

The saliva is an advantageous, non-invasive, and cost-effective biofluid, reflecting both the local (oral) and systemic health status of the human body [24]. Salivary glands are surrounded by capillaries, resulting in a continuous exchange of molecules between blood flow and saliva secretion from the acinus cells. Therefore, the circulating molecules absorbed by salivary glands can affect the saliva composition, mirroring the systemic status [25]. Among the multitude of molecules contained in the saliva, microRNAs (miRNAs) are proving promising biomarkers, representing important epigenetic regulators. Indeed, about 500 of these short non-coding RNA molecules have been identified in saliva [26]. Interestingly, alterations of several miRNAs’ levels have been reported in the maternal circulation depending on pregestational BMI [27], and a few salivary miRNAs have been identified as predictors of periodontal disease in both non-diabetic and diabetic non-pregnant patients [28]. However, to our knowledge, miRNAs’ profiles in the saliva have never been investigated during pregnancy in the presence of obesity and possible related co-morbidities.

This case–control observational study aimed to assess salivary miRNAs levels in a group of obese pregnant women that were previously deeply characterized for periodontal diseases [23].

Moreover, further epigenetic modifications such as gene methylation are also known to be deregulated in obesity and obesity-related insulin resistance. Therefore, methylation levels of specific genes that are well-known players in this context were assessed in maternal saliva: Suppressor Of Cytokine Signalling-3 (*SOCS3*), which is upregulated in response to obesity-related stimuli [29], and Transforming Growth Factor-Beta1 (*TGF-Beta1*), which is a key cytokine in obesity and insulin resistance and has been reported at higher levels in women with a prior history of GDM [30].

## 2. Materials and Methods

### 2.1. Study Participants

This was a pilot study performed on a cohort of pregnant women enrolled in the antenatal clinic at the regular checkup in the Obstetric Unit of the L. Sacco Hospital (ASST Fatebenefratelli-Sacco) in Milan. The study was conducted in accordance with the Declaration of Helsinki and in compliance with current Good Clinical Practice guidelines, local laws, regulations, and organizations. The protocol was approved by the hospital ethical committee (Prot. N. 469/2010/52/AP). Participants gave their written informed consent to collect personal data and biological samples. Pregnant women were enrolled during the first trimester and had a regular clinical follow-up during all trimesters. A deep oral health characterization of this population has been previously reported [23].

Only Caucasian women with a singleton spontaneous pregnancy and aged between 18 and 40 years were enrolled. Exclusion criteria were maternal and fetal infections, fetal malformations, chromosomal disorders, maternal alcohol/drugs abuse, and pregestational Body Mass Index (BMI) < 18.5 or > 25. Detailed inclusion and exclusion criteria are reported in Appendix A.

Thirty-two pregnant women were classified according to their pregestational BMI as Normal-Weight (NW, *n* = 16: 18.5 ≤ BMI ≤ 24.9 kg/m^2^) or Obese (OB, *n* = 16: BMI ≥ 30 kg/m^2^). Ten out of sixteen OB women were diagnosed with Gestational Diabetes Mellitus (GDM), based on oral glucose tolerance test (75 g), according to our clinical protocol. NW women had uncomplicated pregnancies. Therefore, GDM was an exclusion criterion in this group.

OB patients were followed up in a dedicated antenatal clinic, providing nutritional counseling on diet and lifestyle, particularly focused on pregnancy weight gain. Obese patients with GDM received further advice to control their glycemia according to the clinical protocol.

### 2.2. Clinical Data and Biological Samples Collection

*Clinical data:* Maternal, neonatal, and placental data were recorded at recruitment and after delivery.

*Oral and periodontal health:* A complete clinical oral health examination was performed in the third trimester, as previously reported [23,31]. Pregnant patients were classified as *healthy* with an overall good oral care (no dental soft debris—dental plaque or materia alba or calculus—or nor gingival Bleeding On Probing (BOP) or Probing Pocket Depth (PPD > 3 mm), or *periodontally diseased* with an overall neglected oral care, which grouped *gingivitis* (soft debris and calculus in ≥ 6 teeth and gingival bleeding ≤ 5 teeth) and *periodontitis* (probing pocket depth ≥ 4 mm, soft debris, calculus, and bleeding ≥ 6 teeth).

*Saliva collection:* Saliva sampling was performed during the third trimester without any stimulation (passive drool technique), thus avoiding the use of solutions that could alter physico-chemical features and preventing the employment of tools requiring a specific site of withdrawal. Choosing the same timing of withdrawal for all subjects eludes any fluctuations in the saliva composition related to circadian rhythms. Enrolled women were asked to refrain from eating, drinking sugar or alcoholic beverages, smoking, and performing invasive oral care procedures for at least 1h before collection. They were then asked to rinse their mouth (1 min) with a physiological solution to remove any food residues, which could alter pH or promote bacterial growth, and afterward, not to swallow for 3 min and spit in a sterile tube. Collected samples were temporarily stored at −80 °C.

### 2.3. Isolation of Extracellular Vesicles and miRNA

Saliva samples (1 mL) were thawed on ice to avoid Extracellular Vesicles’ (EVs) thermal damage. Then, they were centrifuged at 4000× *g* for 30 min at 4 °C to remove any cell debris and aggregates. Supernatants were ultracentrifuged at 110,000× *g* for 75 min at 4 °C in order to pellet EVs, then stored at −20 °C.

miRNAs’ isolation from the obtained EVs was performed with the combination of the miRNeasy Kit and RNeasy Cleanup Kit (Qiagen, Hilden, Germany), according to the manufacturer’s protocol. They were eluted in 20 µL of nuclease-free water and stored at −80 °C until used. EV miRNAs’ quality and integrity were assessed through the “2100 Bioanalyzer RNA system” with the Pico Kit (Agilent Technologies, Santa Clara, CA, USA), but the concentration (ng/µL) was assessed by a Quantus Fluorometer (Promega, Milan, Italy).

### 2.4. Screening of miRNA Expression

There were 15 ng miRNAs Reverse Transcribed (RT), preamplified (16 cycles), and analyzed by real-time PCR with the QuantStudio™ 12K Flex OpenArray^®^ Platform (Applied Biosystems, Waltham, MA, USA), as previously described in detail [32]. Gene Expression Suite Software v.1.0.3 (Applied Biosystems, Waltham, MA, USA) was used to process miRNA profiling expression data from the “TaqMan™ OpenArray™ human MicroRNA panel” (ThermoFisher Scientific, Waltham, MA, USA). As described in Figure 1, we obtained 758 Relative Threshold Cycle (Crt) values for each subject, which included 754 unique miRNAs and four internal controls (ath-miR159a, RNU48, RNU44, and U6). For each amplification curve, we obtained an AmpScore value, a quality measurement that indicates the low signal in the amplification curve linear phase (range: 0–2). miRNAs with Crt value > 28, or AmpScore < 1.24, or missing were considered not amplified, and thus, the corresponding Crt value was set to 29. miRNAs that were not amplified in all subjects (*n* = 420) were excluded, resulting in 334 miRNAs being included in the statistical analysis. miRNA expression was determined using the relative quantification 2-ΔΔCrt [33]. Among the endogenous controls analyzed (i.e., ath-miR159a, U6-rRNA, RNU44, RNU48), U6-rRNA and RNU48 were chosen for normalization by the NormFinder algorithm [34], due to their stability in the comparison between samples.

### 2.5. DNA Isolation

Total DNA was extracted using the QIAamp^®^ DNA Mini and Blood Mini Kit (Qiagen; Valencia, CA, USA) following the supplementary protocol (“Isolation of genomic DNA from saliva and mouthwash using the QIAamp^®^ DNA Blood Mini Kit”). Briefly, saliva (160 μL) was thawed on ice and added with 640 μL of Phosphate Buffered Saline (PBS). Samples were then centrifuged at 1800× *g* for 10 min and their supernatant was discarded. The pellets were lysed and DNA extracted with a spin-column-based method. DNA was eluted in 44 μL of Buffer AE (10 mM Tris·Cl; 0.5 mM EDTA; pH 9.0) for a final concentration of 25 ng/μL.

### 2.6. Bisulfite Conversion and DNA Methylation

500 ng di genomic DNA was bisulfite converted using the EZ DNA Methylation Direct Kit (Zymo Research, Orange, CA, USA), in accordance with the manufacturer’s protocol. Converted DNA was eluted in 30 μl of Elution Buffer.

Analysis of DNA methylation was performed by PyroMark MD Pyrosequencing System (Qiagen, Milan, Italy) as previously described [35,36].

The bisulfite-treated genomic DNA samples were amplified with PCR; CpG sites were analyzed within the promoter regions of the following genes: Suppressor of Cytokine Signalling 3 (*SOCS3*) and Transforming Growth Factor Beta 1 (*TGF-Beta1*).

Detailed information concerning primer sequences and genomic regions is listed in Table 1. The percentage of 5- methylcytosine (% 5mC) was reported as the percentage of methylated cytosine divided by the sum of methylated and unmethylated cytosines. Every sample was tested twice for each marker to confirm reproducibility and to increase the precision of the findings.

### 2.7. Statistical Analysis and Prediction Tools

Maternal, neonatal, and placental characteristics and periodontal health parameters were compared between groups by the Student *t*-test or Mann–Whitney U test for independence samples according to the data distribution (assessed by the Kolmogorov–Smirnov test). Differences between groups in the frequencies of periodontal disease, mode of delivery, neonatal weight centile, and fetal sex were evaluated by the chi-squared test for independence or Fisher’s exact test.

Multivariable linear regression models were applied to verify the association between BMI groups and miRNA expression in the screening phases. miRNA expression values were log2-transformed to achieve a normal distribution. Regression models were adjusted for age, smoking habit, GDM, and gestational age at saliva withdrawal. Due to the high number of comparisons, we applied a multiple comparison correction method based on the Benjamini–Hochberg False Discovery Rate (FDR) to calculate the FDR *p*-value. The criteria used to identify the top miRNAs were a *p*-value < 0.05, an FDR *p*-value < 0.20, and a Fold Change (FC) < 0.5 or FC > 2. A Volcano plot was produced to select miRNAs characterized by more than ±2-fold case–control differences (FC 2 or < 0.5) with a *p*-value < 0.05 from linear regression models.

To evaluate the associations between methylation levels (*SOCS3*, *TGF-Beta1*) and BMI groups, we used multiple linear regression models adjusted for maternal age, smoking habits, and GDM. We report marginal means and 95% CI.

Statistical analyses were performed using the statistical package SPSS, v.27 (IBM, Armonk, NY, USA) and SAS 9.4 statistical software (SAS Institute Inc., Cary, NC, USA).

*miRNA pathway enrichment analysis:* miTALOS v2 tools (http://mips.helmholtz-muenchen.de/mitalos (accessed on 15 March 2022)) were used to gain insights into tissue-specific miRNA regulation of biological pathways. Indeed, miTALOS integrates five different miRNA target prediction tools and two different resources (KEGG and NCI), being able to identify a specific association between miRNAs and signaling pathways [37].

We integrated our analysis by using DIANA–miRPath v3 (https://dianalab.e-ce.uth.gr/html/mirpathv3/index.php?r=mirpath (accessed on 15 March 2022)), which uses data from the KEGG and TarBase resources, to perform miRNA pathway enrichment analysis. We thus investigated how these pathways are combined in a posteriori bioinformatic analysis (union and intersection of pathways search). The results are shown as miRNA/pathways’ interaction heat maps [38].

For both miTALOS v2 and DIANA–miRPath v3, the significance of the association (*p*-value) between miRNAs and the resulting pathways was calculated with Fisher’s exact test; then, the results for multiple pathways were corrected using the EASE score and false discovery rate (Benjamini–Hochberg) [39,40,41].

## 3. Results

### 3.1. Maternal and Oral Health Characteristics and Delivery Data

Table 2 resumes the maternal and oral health characteristics.

Pregestational BMI was significantly higher in the OB group compared to NW (*p* < 0.001), according to the inclusion criteria. Gestational Weight Gain (GWG) was significantly lower in OB vs. NW (*p* = 0.035), in compliance with IOM indications (upper limits: +9 kg for OB, +16 kg for NW). However, OB women gained more overall excessive weight than recommended for OB, as shown by the percentages reported in Table 2 (maternal GWG to IOM upper limit). Maternal basal glycemia was higher in the OB group (*p* = 0.034).

Unstimulated saliva was collected at a similar gestational age, and its flow rate had a normal range in both groups [42].

Concerning the overall oral status, a higher percentage of mothers with periodontal disease was observed in the OB group (68.8%) compared to NW (56.2%), though not significantly. Among the recorded measures of dental and periodontal health, the number of teeth was significantly lower (*p* = 0.027) in the OB vs. NW group. The dental plaque index percentage was significantly higher in OB women vs. NW (*p* = 0.024).

All the enrolled women delivered at term, without obstetrical complications. No difference was recorded in the mode of delivery, although a higher percentage of cesarean section was observed in the OB group (NW: 12.5%; OB: 43.8%), in agreement with other epidemiological data from obese pregnancies.

Table 3 reports neonatal and placental data at delivery.

Neonates were similar in terms of gestational age and biometric parameters (weight and centiles); as expected, a higher number, though not significant, of Large for Gestational Age (LGA) babies was observed from OB mothers. Fetal sex frequencies did not differ between the two study groups.

Placentas were significantly heavier (*p* = 0.035) and thicker (*p* = 0.050) in the OB than in the NW group.

### 3.2. miRNAs’ Profile in Maternal Saliva

In maternal saliva, the presence of 754 miRNAs was evaluated by a simultaneous quantitation with the *TaqMan OpenArray Human MicroRNA Panel*, which contains the most biologically relevant miRNAs in miRBase (https://www.mirbase.org/, accessed on 1 March 2011). Data were adjusted for gestational age at saliva withdrawal, maternal age, and smoking habits, as well as GDM association as covariates.

The expression of 334 miRNAs was detectable in the saliva of at least one subject of our study population. Among these miRNAs, 31 were differentially expressed (FC < 0.5 or > 2.0) with statistical significance (*p* < 0.05) between OB and NW mothers, as shown by the volcano plot (Figure 2).

Following FDR correction, 20 miRNAs were accepted for the downstream analysis (FDR < 0.20). In particular, 2 miRNAs out of 20 (hsa-miR-505 and hsa-miR616) had FC < 0.5, thus resulting in being downregulated in the OB group. FC > 2.0 indicates upregulated levels for 18 miRNAs in OB women. FC, *p*-value, and FDR are summarized in Table 4.

To gain insights into the miRNA profiling results, miTALOS and DIANA–miRPath were used for miRNA pathway enrichment analysis.

First, all 20 miRNAs were inserted together in miTALOS. Interestingly, the union analysis (i.e., combinations of pathways related to the inserted miRNAs) showed a significant association between the 20 miRNAs and a specific pathway involving *TGF-Beta* signaling (*p* = 0.040).

An additional analysis was performed by DIANA–miRPath v3. A heat map depicting the force of association was obtained after inserting the 20 miRNAs differentially expressed in OB vs. NW maternal saliva. Four pathways matches were pointed out by the DIANA–miRPath pathway enrichment analysis (Figure 3 and Table 5).

### 3.3. DNA Methylation in Maternal Saliva: TGF-Beta1 and SOCS3

Methylation data were corrected for maternal age, smoking habits, and GDM association. Both *TGF-Beta1* and *SOCS3* were differentially methylated, with significantly decreased methylation levels in OB mothers compared to NW (*p* = 0.019 and *p* = 0.025, respectively) (Table 6).

## 4. Discussion

Maternal obesity and Gestational Diabetes (GDM) have been associated with lipotoxicity, inflammation, and increased oxidative stress both at systemic and intrauterine levels [8,10,44]. We previously reported placental modifications in obese (OB) pregnancies, indicating an excessive oxidative intrauterine/systemic environment [13,15,45]. Increased antioxidant capacity and C-reactive protein levels in both maternal plasma and the saliva of obese women with and without GDM have also been reported, confirming saliva as an effective fluid mirroring the systemic status, representing an interesting non-invasive tool for the evaluation of metabolic signatures during pregnancy [23].

The obesogenic environment can influence pregnancy outcomes via epigenetic mechanisms, including microRNAs (miRNAs) and DNA methylation [46,47]. Alterations occurring in maternal obesity and GDM can affect epigenetic modifications of different maternal and fetal tissues, leading to alterations in several cellular pathways, which can affect the future development in the offspring of obesity, diabetes, and other metabolic and cardiovascular diseases [3,47,48,49]. Studies on maternal circulating miRNAs are currently in progress aiming at identifying specific miRNA profiles associated with pregnancy pathologies [50,51,52,53], some of them proposing specific miRNAs as early biomarkers for gestational diseases such as preeclampsia [50], obesity [51,52], or GDM [53]. Nevertheless, to our knowledge, this is the first study characterizing the miRNAs’ profile and DNA methylation in the saliva of obese compared to normal-weight pregnant women.

Mothers included in this study population had previously been carefully classified for periodontal comorbidities (gingivitis and periodontitis) [23]. Clinical conditions of the enrolled patients were well characterized during regular prenatal checks, with obese women receiving specific nutritional and lifestyle counseling in a dedicated clinic. Indeed, OB gained significantly less weight during pregnancy compared to normal-weight mothers. Nevertheless, obese women exceeded gestational weight gain IOM recommendations, possibly worsening the obesity-related adverse systemic and local environment. In fact, the poorest overall periodontal health status was observed in the OB group. Moreover, both placental weight and thickness were increased in OB, suggesting a trend to a lower placental efficiency, giving rise to a higher, though not significant, mean neonatal weight centile. This obesogenic context also resulted in alterations of the saliva epigenetic profile. In particular, we reported a set of 20 up regulated or down expressed miRNAs and decreased methylation levels of the genes *SOCS3* and *TGF-Beta1* in the saliva of obese vs. normal-weight mothers.

### 4.1. miRNAs’ Profile

The saliva shows a very high number of detectable miRNAs, having a unique spectrum of these non-coding RNAs [54].

We performed a pathway enrichment analysis by integrating different miRNA target prediction tools, to identify specific associations with signaling pathways. In particular, saliva miRNAs showing significant differences between OB and NW mothers, matched pathways involving fatty acids biosynthesis and metabolism, Extracellular Matrix (ECM)–receptor interaction, and lysine degradation.

#### 4.1.1. Fatty Acids Biosynthesis and Metabolism

We found significant associations between miR-1254 and fatty acids biosynthesis and metabolism. Interestingly, our group recently reported lower levels of Long Chain-Polyunsaturated Fatty Acid (LC-PUFA) derivatives, arachidonic acid, and DHA, opposite significantly increased saturated palmitic acid levels, in obese placentas [15]. These data can indicate a disruption of the physiologic LC-PUFA biomagnification linked to maternal obesity, similar to alterations occurring in pregnancy pathologies characterized by placental dysfunction, such as intrauterine growth restriction. Moreover, changes in fatty acids biosynthesis and metabolism can damage cells’ and organelles’ membranes, leading to excessive ROS production and oxidative stress, characterizing the systemic and intrauterine environment of obese and GDM pregnancies [3,13,44]. Although this evidence is derived from different tissues, the present results may be suggestive of a disarranged fatty acid metabolism in obese pregnancies mediated by miRNA epigenetic alterations, which can also be detected in the oral non-invasive fluid during the third trimester of pregnancy.

#### 4.1.2. Extracellular Matrix

Pathway enrichment analysis showed a significant match between hsa-miR-184, hsa-miR-206, hsa-miR-302c-5p, hsa-miR-376a-3p, hsa-miR-505-3p, hsa-miR-512-3p hsa-miR-635, and the ECM–receptor interaction pathway.

Intriguingly, human endothelial cell culture exposed to high levels of saturated fatty acids showed alterations in the structure and functionality of the components of the ECM [55]. Similarly, hyperglycemia induces excessive production of ECM collagens and altered proteoglycans by endothelial cells. Indeed, also in patients with type II diabetes, the ECM of blood vessels presented an altered ratio of proteoglycans and a thicker basement membrane vs. normoglycemic controls. These changes could represent pathogenic mechanisms for vascular diabetic complications, which have a strong inflammatory basis [56]. Physiologically, ECM remodeling consists of structural and compositional changes and is critical for the differentiation and expansion of adipocytes. In obesity-related adipose tissue hypertrophy, ECM synthesis and degradation can be altered, causing inflammation and mechanical stress [57]. Both ECM degradation and pro-inflammatory milieu establishment are promoted by the activity of Matrix Metalloproteinases (MMPs), which are well-known targets of several miRNAs, suggesting their epigenetic regulation [58]. In turn, pro-inflammatory cytokines and pathogens associated with periodontal disease raise MMPs’ secretion, thus favoring disintegration of the ECM [59]. Therefore, the association between ECM pathways and OB-related salivary miRNAs’ alterations reported in the present study might be speculated to be part of the dysregulation of the ECM remodeling, due to the obesogenic environment, the linked inflammatory setting, and the oral disease condition.

#### 4.1.3. Lysine Degradation

hsa-miR-302d-3p, hsa-miR-302c-3p, hsa-miR-376a-5p, hsa-miR-505-3p, hsa-miR-505-5p, hsa-miR-512-3p, and hsa-miR-616-3p resulted in being associated with the lysine degradation pathway.

Amino Acids (AAs) are necessary building blocks for protein biosynthesis. Lysine is crucial for cell growth and plays a fundamental role in the production of carnitine, which shuttles long-chain fatty acids into mitochondria for energy production and assists in lowering cholesterol levels [60].

Coherent with the results reported in this manuscript, decreased lysine levels in the adipose tissue of obese non-pregnant subjects have been shown [61], suggesting an altered metabolic pattern typical of the obese condition. Desert and colleagues reported increased lysine levels in the urinary excretion of obese mothers vs. normal-weight, hypothesizing an enhancement in lysine renal secretion [62]. In a metabolomics analysis of OB vs. NW placentas, we previously showed amino acid changes, with marked lower amounts of lysine [15]. Similar amino acid alterations were previously reported in the serum of hyperglycemic mothers, showing a metabolic profile consistent with insulin resistance [63,64].

Despite the consistency of this evidence, the underlying mechanism explaining the relation between obesity and lysine catabolism is still unknown. It is tempting to hypothesize that miRNAs could be involved in these mechanisms. Indeed, the lysine degradation pathway has been shown as a possible target of specific miRNAs related to a diabetic microenvironment in a beta-human cell line [65]. Targeted studies are needed to confirm this hypothesis.

### 4.2. DNA Methylation

#### 4.2.1. TGF-Beta1

Transforming Growth Factor-Beta 1 (TGF-Beta1) is an anti-inflammatory cytokine with widespread and multi-organ effects. It plays an active role in glucose, lipid, amino acid, and redox metabolism and represents a key cytokine in obesity and insulin resistance [30,66,67,68]. Noteworthy evidence showed that TGF-Beta1 signaling inhibition ameliorated the metabolic profile in mice by increasing glucose and insulin tolerance. In turn, elevated TGF-Beta1 levels were associated with glucose intolerance and increased adiposity [69] and have been reported in women with a prior history of GDM [70]. Accordingly, in the present study, *TGF-Beta1* methylation levels were significantly decreased in the saliva of obese vs. lean mothers, supporting its upregulation, even when adjusted for maternal age, smoking habits, and the presence of GDM. It might be hypothesized that OB’s *TGF-Beta1* higher systemic levels are mirrored in the saliva, confirming this biofluid as a striking non-invasive tool for the investigation of local and systemic alterations. Moreover, the pathway enrichment analysis by miTALOS showed a noticeable significant association between the 20 miRNAs differentially expressed in OB and the *TGF-Beta* signaling pathway. This evidence supports previous results showing a miRNA-based regulation of *TGF-Beta* signaling [71] and represents an interesting starting point to unravel the alterations of the molecular regulatory mechanisms of this pathway in obesity, leading to the possible impairment of nutrients and redox metabolism, with the consequent promotion of oxidative stress.

#### 4.2.2. SOCS3

Over the past few years, SOCS3 has emerged as an interesting target to treat metabolic disorders. Indeed, SOCS3 association with inflammation, cumulative stress, and insulin resistance in obesity and diabetes is well established, and its inhibition is considered a promising strategy for the treatment of metabolic disorders in different conditions [29]. In pregnancy, the possible participation of SOCS3 in the attenuation of leptin and insulin signaling has been recently reported, with increased resistance associated with SOCS3 upregulation during physiological pregnancy-induced metabolic changes [72].

Here, we report a significant decrease in *SOCS3* methylation levels in the saliva of OB women compared to NW, likely accounting for exacerbated *SOCS3* expression in pregnant women with higher BMI. In non-pregnant individuals, *SOCS3* DNA methylation has been previously shown to be inversely related to body weight and to modulate the impact of overall stress on obesity [73]. Furthermore, obesity-related inflammation leads to the upregulation of SOCS3 proteins in several tissues, leading to the inhibition of insulin signaling and, therefore, representing an interesting mechanism connecting the immune and metabolic system in a delicate and balanced cross-talk [29,74]. Results reported in the present study suggest a disruption of this balance, and the altered *SOCS3* DNA methylation may represent an intriguing mechanism undergoing OB metabolic alterations, deserving further future investigations both as a biomarker of the dysmetabolic context in pregnancy and as a possible future target for its treatment.

### 4.3. Strengths and Limitations

This work is a pilot study, therefore presenting a small sample size. Nevertheless, the analyzed population was very well defined in terms of inclusion and exclusion criteria, clinical characteristics, and associated conditions, which strongly reduced any additional clinical bias impacting the results. The strict statistical analysis that was applied included several covariates, allowing us to focus on the effect of obesity on the analyzed epigenetic modifications.

Molecular investigations were performed on the saliva. Indeed, this is an advantageous, non-invasive, and cost-effective biofluid. Saliva composition reflects perturbations occurring in the oral cavity, which might vary depending on the local conditions. However, it can also mirror the systemic status, representing an intriguing tool for biomarker non-invasive examination. Moreover, a careful procedure was employed to remove any contaminants and to avoid chemical alterations or the release of inflammatory mediators. Indeed, the passive drool technique for saliva collection is recommended when the target concentration is low, and the unstimulated collection allows obtaining high purity and great sample quality [54]. 

## 5. Conclusions

To our knowledge, this is the first study exploring the epigenetic status in the saliva of obese pregnant women. This pilot study can pave the way for future studies on pregnancy-related epigenetics. Among the mechanisms undergoing obese-related injuries, epigenetic modifications have gained increasing attention over the past years for their high potential as both disease biomarkers and treatment targets. We characterized the epigenetic signatures of a group of clinically well-characterized obese and normal-weight pregnant women by analyzing their saliva, which represents a promising non-invasive biofluid, which can mirror the systemic status of the individual. We found alterations in the expression of 20 different miRNAs and methylation levels of two genes involved in obesity-related inflammation and oxidative stress. These results help to clarify some of the mechanisms involved in the impairment occurring in obesity, which could contribute to the increase in the risk of negative outcomes in OB pregnancies. Furthermore, the underlying association between miRNAs and their related molecular pathways will help understanding specific damaged pathways. Since miRNAs could be promising biomarkers in the early diagnosis and treatment of pregnancy-adverse conditions, these results place the analyzed salivary miRNAs in a clinical context for future studies and propose possible molecular mechanisms in which they could be involved, representing future promising tools for the identification of obesity-related alterations. Further studies are needed for investigating maternal miRNA profiling related to post-natal follow-up data.

## Figures and Tables

**Figure 1 nutrients-14-02122-f001:**
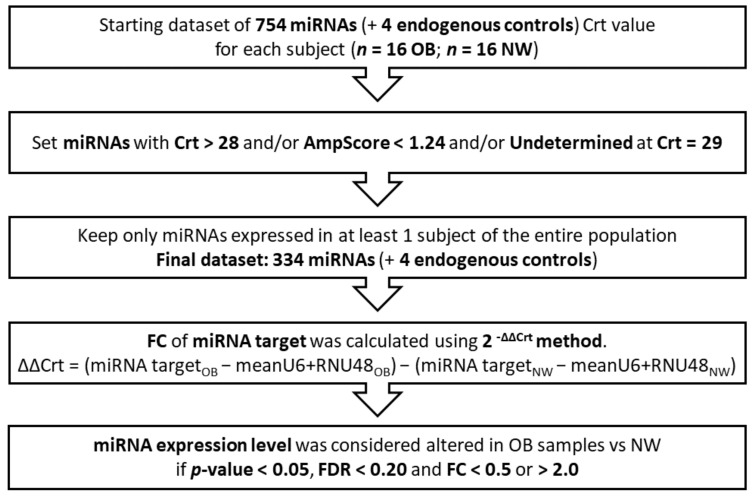
Data processing workflow. OB: Obese, NW: Normal-Weight, Crt: Relative Threshold Cycle, FDR: False Discovery Rate; FC: Fold Change; Bold: highlights the most relevant terms.

**Figure 2 nutrients-14-02122-f002:**
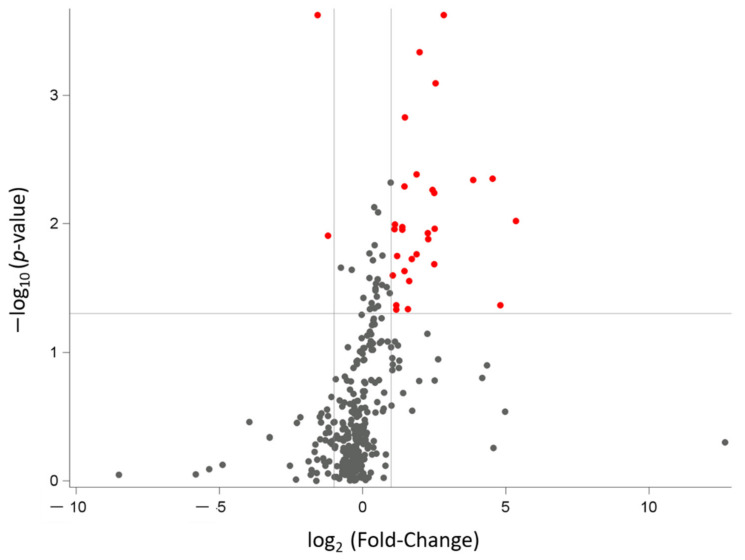
Volcano plot showing differentially expressed miRNAs between OB and NW women. The graph reports the 334 detectable miRNAs (all black and red dots). The horizontal line states the –log_10_ of the *p*-value 0.05; the two vertical lines indicate the log_2_ of the FC values 0.5 and 2.0. As a result, miRNAs with a significantly different expression vs. NW are displayed in red (*n* = 31). In particular, the upper left quadrant reports the downregulated miRNAs, while the upper right one all the upregulated miRNAs.

**Figure 3 nutrients-14-02122-f003:**
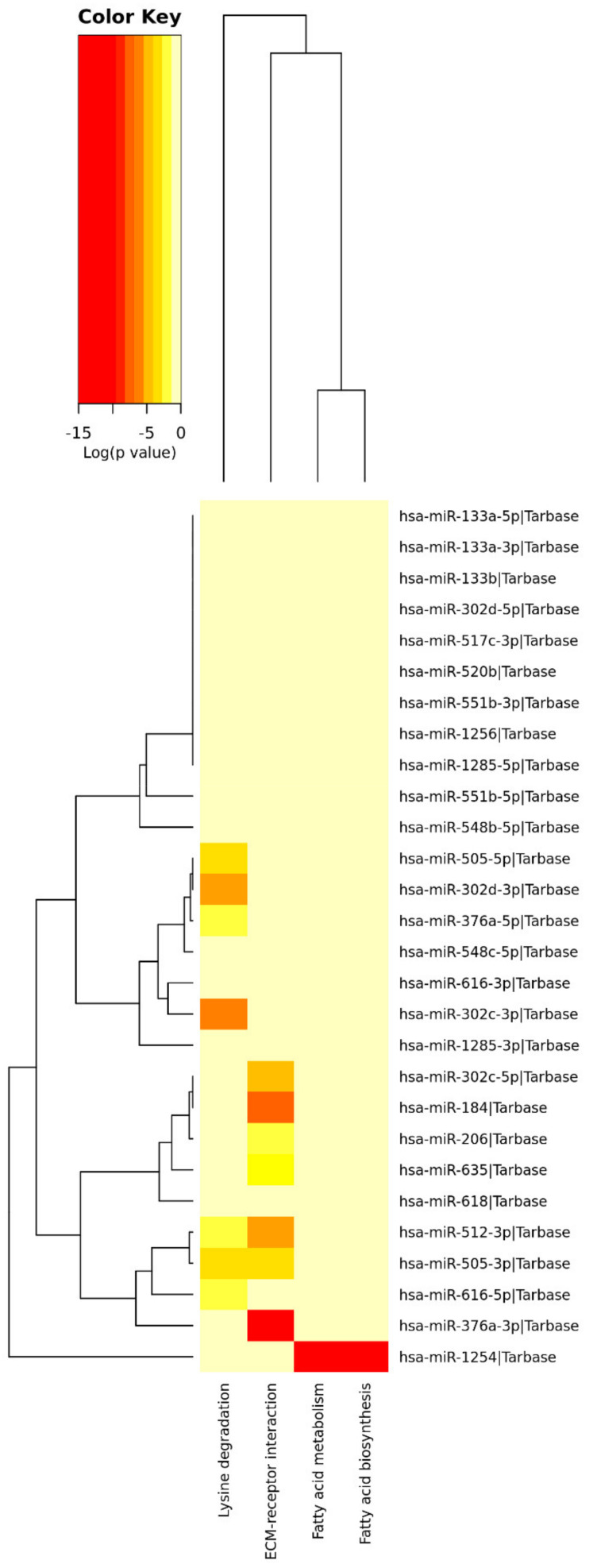
Heat map by DIANA–miRPath v.3. The analyzed miRNAs are listed on the right. The 4 specific pathways, resulting in being significant from miRNA pathway enrichment analysis, are listed at the bottom. The force of association (Log(*p*-value)) between the analyzed miRNAs and the 4 pathways is indicated by the color key reported at the upper left corner. ECM: ExtraCellular Matrix, Log: log_10_
*p*-value.

**Table 1 nutrients-14-02122-t001:** Pyrosequencing assays information.

Gene	Chromosome Position	CpG Sites	Primers: Forward (F) Reverse (R) Sequencing (S)	Sequencing Length (BP)	Annealing Temperature (°C)
** *TGF-Beta1* **	chr17:6354421-76354821	2	F: TTGGGTGATTTTTTTATAGGAGTT R: bio-TCCCCCCAAAAAAACCTATT S: GAGATGTTGAAGAGTGGTTA	25	52
** *SOCS3* **	chr19:41353024-41353458	2	F: GGTTTGTTTTTTGAGTTTT R:bio-CTACAAAAACTAAAAATCTCCC S: TATTTATTTTTTGGTATTAG	23	54

BP: Base Pairs, °C: Celsius degrees.

**Table 2 nutrients-14-02122-t002:** Maternal and periodontal health characteristics.

	NW *n* = 16	All OB *n* = 16	*p*-Value
Age, years ^B^	30.94 ± 3.89	33.25 ± 4.61	ns
Pregestational BMI, kg/m^2 B^	21.02 ± 2.25	36.47 ± 5.53	<0.001
GWG, kg ^B^	13.38 ± 4.41	9.13 ± 5.63	0.035
GWG to IOM upper limit, % ^A^	83.61 ± 27.54	101.37 ± 62.57	ns
Basal glycemia, mg/dL ^B^	81.50 ± 5.48	92.69 ± 15.18	0.034
Gestational diabetes, *n* (%)	0 (0)	10 (62.5)	-
GA at saliva withdrawal, mL ^A^	33.24 ± 1.73	32.99 ± 2.52	ns
Saliva flow rate, mL/min ^A^	0.48 ± 0.22	0.47 ± 0.17	ns
Periodontal health ^C^			
Healthy, n (%)	7 (43.8)	5 (31.2)	ns
Periodontal disease, n (%)	9 (56.2)	11 (68.8)	
Number of teeth ^B^	27.60 ± 0.74	25.31 ± 3.01	0.027
BOP, % sites ^B^	17.98 ± 19.76	39.57 ± 38.10	ns
PPD, mean ^A^	2.38 ± 0.45	2.63 ± 0.88	ns
Plaque index, % ^A^	25.61 ± 18.97	52.90 ± 40.32	0.024
Calculus, % ^A^	27.79 ± 23.65	44.53 ± 35.81	ns

Values expressed as the mean ± standard deviation were analyzed according to their distribution with independent samples ^A^: Student *t*-test, or ^B^: Mann–Whitney U test, or ^C^: chi-squared test for independence or Fisher’s exact test; statistical significance vs. normal-weight women. BMI: Body Mass Index; GWG: Gestational Weight Gain at delivery; IOM: Institute of Medicine; maternal basal glycemia: maternal fasting glycemia refers to the first value of the Oral Glucose Tolerance Test (OGTT; physiological value ≤ 92 mg/dL) performed between 24 and 28 weeks of gestation; GA: Gestational Age; flow rate: ratio between mL of saliva and minutes of withdrawal; oral disease: gingivitis and/or periodontitis; PPD: Probing Pocket Depth; BOP: gingival Bleeding On Probing; ns: not statistically significant.

**Table 3 nutrients-14-02122-t003:** Neonatal and placental data at delivery.

	NW *n* = 16	All OB *n* = 16	*p*-Value
GA at delivery, wks ^A^	39.76 ± 0.87	39.33 ± 1.18	ns
Neonatal weight, gr ^A^	3331.25 ± 294.60	3488.75 ± 356.62	ns
Neonatal weight centile ^A^	47.56 ± 24.19	61.00 ± 28.72	ns
AGA, *n* (%) ^C^	15 (93.8)	11 (68.8)	ns
LGA, *n* (%) ^C^	1 (6.3)	5 (31.3)	
Neonatal sex, *n* ^C^			ns
Males, *n* (%)	9 (56.3)	7 (43.8)	
Females, *n* (%)	7 (43.8)	9 (56.3)	
Placental weight, gr ^A^	439.00 ± 79.05	509.29 ± 80.81	0.035
Neonatal/placental weight ^B^	7.78 ± 1.66	6.97 ± 1.63	ns
Placental area, cm^2 B^	281.94 ± 64.06	266.16 ± 45.33	ns
Placental thickness, cm ^A^	1.61 ± 0.37	1.97 ± 0.50	0.050

Values are expressed as the mean ± standard deviation. Data were analyzed according to their distribution with independent samples ^A^: Student *t*-test, or ^B^: Mann–Whitney U test, or ^C^: chi-squared test for Independence or Fisher’s exact test; statistical significance vs. normal-weight women. GA: Gestational Age; Neonatal Weight Centile: calculated using INeS Charts (http://www.inescharts.com/index.aspx (accessed on 15 March 2022)); refer to [43]; AGA: Appropriate for Gestational Age; LGA: Large for Gestational Age.

**Table 4 nutrients-14-02122-t004:** Upregulated and downregulated miRNAs in maternal saliva.

miRNA Name	Fold Change (FC)	*p*-Value	False Discovery Rate *p*-Value (FDR)
hsa-miR-505 ↓	0.335	0.0002	0.0396
hsa-miR-616 ↓	0.434	0.0123	0.1873
hsa-miR-618 ↑	7.141	0.0002	0.0396
hsa-miR-206 ↑	3.985	0.0005	0.0514
hsa-miR-376a ↑	5.870	0.0008	0.0671
hsa-miR-517c ↑	2.788	0.0015	0.0989
hsa-miR-133a ↑	3.703	0.0041	0.1599
hsa-miR-512 ↑	23.295	0.0044	0.1599
hsa-miR-302d ↑	14.427	0.0046	0.1599
hsa-miR-520b ↑	2.755	0.0051	0.1599
hsa-miR-1254 ↑	5.443	0.0054	0.1599
hsa-miR-133b ↑	5.634	0.0057	0.1599
hsa-miR-1285 ↑	40.956	0.0095	0.1859
hsa-miR-635 ↑	2.178	0.0102	0.1859
hsa-miR-551b ↑	2.614	0.0106	0.1859
hsa-miR-548b-5p ↑	5.692	0.0110	0.1859
hsa-miR-1256 ↑	2.169	0.0110	0.1859
hsa-miR-302c ↑	2.622	0.0111	0.1859
hsa-miR-184 ↑	4.817	0.0118	0.1873
hsa-miR-548c-5p ↑	4.912	0.0131	0.1909

FC (Fold Change), *p*-value, and FDR (False Discovery Rate) of the 20 miRNAs with significantly different expression levels in OB vs. NW. Data were analyzed with a multiple linear regression analysis adjusted for gestational age at saliva withdrawal, age, smoking habits, and GDM association. Results are presented in descending order of statistical significance. (↓: downregulated; ↑: upregulated miRNA in OB vs. NW).

**Table 5 nutrients-14-02122-t005:** Union analysis performed with DIANA–miRPath v.3.

	Pathways Information	*p*-Value	miRNAs
Fatty acids biosynthesis	Synthesis of fatty acids. Fatty acids are generally excessive in the Western diet.	2.1444 × 10^−11^	hsa-miR-1254
Fatty acids metabolism	Anabolic and catabolic processes involving fatty acids or related molecules. Fatty acids are generally excessive in the Western diet.	0.0002	hsa-miR-1254
Lysine degradation	Catabolism of lysine (from dietary up-taken or intracellular proteins). Lysine is generally excessive in the Western diet.	1.7153 × 10^−7^	hsa-miR-505-5p; hsa-miR-302d-3p; hsa-miR-376a-5p; hsa-miR-302c-3p; hsa-miR-512-3p; hsa-miR-505-3p; hsa-miR-616-3p
ECM–receptor interaction	Tissue and organ morphogenesis; maintenance of cell and tissue structure and function; adhesion, migration, differentiation, proliferation, and apoptosis; force-transmitting physical link with the cytoskeleton.	<1 × 10^−325^	hsa-miR-302c-5p; hsa-miR-184; hsa-miR-206; hsa-miR-635; hsa-miR-512-3p; hsa-miR-505-3p; hsa-miR-376a-3p

ECM: ExtraCellular Matrix.

**Table 6 nutrients-14-02122-t006:** *TGF-Beta1* and *SOCS3* methylation levels.

	Mean OB (95% CI)	Mean NW (95% CI)	*p*-Value
*TGF-Beta1* (% 5mC)	0.44 (0.24–0.63)	0.86 (0.61–1.11)	0.019
*SOCS3* (% 5mC)	60.95 (57.20–64.70)	68.50 (63.75–73.25)	0.025

Results from a multiple logistic regression analysis adjusted for age, smoking habits, and GDM association. Data are shown as the mean with the 95% Confidence Interval (CI) of methylated cytosines’ percentage (% 5mC); *p* < 0.05 vs. NW. *TGF-Beta1*: Transforming Growth Factor-Beta 1; *SOCS3*: Suppressor of Cytokine Signalling 3.

## Data Availability

All data that support the findings of this study are available from the corresponding author (Chiara Mandò) upon reasonable request.

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
