# Peer review of "Epigenetic Profiling in the Saliva of Obese Pregnant Women"

_nutrients, 2022, doi:10.3390/nu14102122_

Round 1

Reviewer 1 Report

Thank you for the opportunity to review this manuscript submitted to nutrients, it is extremely well written well conceived and well executed I have no significant complaints or critiques to the authors

Author Response

We thank the reviewer for this gratifying comment.

Reviewer 2 Report

Overall, this is a well written article. However, the authors would have to describe the unique contribution of this study to the current literature.

Author Response

We thank the reviewer for his/her positive feedback and constructive suggestion.

Indeed, to our knowledge, this is the first study exploring the epigenetic status in the saliva of obese pregnant women. We believe that this pilot study could pave the way for future studies on pregnancy-related epigenetics, especially using saliva as innovative and non-invasive biofluid for the identification of novel epigenetic biomarkers.

We now added a comment about it in the conclusion section (lines 489-491).

Reviewer 3 Report

Dear authors,

Thanks for sharing your work, in general, I think you have valuable data and I recommend publication, even if some revision is necessary from my point of view as a clinical researcher.

Here are some suggestions and requests for improvements:

Firstly, I recommend adding follow-up results of the participants. As you mentioned in the materials and method section, the patients were followed-up in antenatal clinics. This result will aid your conclusion; “The characterization of the epigenetic profile in the saliva of pregnant women can represent a promising tool for the identification of obesity related altered mechanisms and of possible biomarkers for early diagnosis and treatment of pregnancy adverse conditions.”

Secondly, I recommend adding an inclusion-exclusion chart as a figure 1 or a supplementary figure. Readers may ask how many original enrollments and exclusions happened. For example, you can see the figure 1 of the other Nutrients paper: https://doi.org/10.3390/nu13082825

Thirdly, table 2 can be augmented with the percentage of underlying or previous disease history. For example, hyperlipidemia 6 (30%) vs 9 (50%) or so. (If you have the information's on underlying disease characteristics)

As a minor revision, I also would like to say in line 128, “and pregestational BMI <18.5 or between 25 and 30.”, could be more simplified as “<18.5 or >25.”

Thanks for your work to improve our knowledge and the Nutrients. Have a nice day.
